# Using ship-borne observations of methane isotopic ratio in the Arctic Ocean to understand methane sources in the Arctic

Berchet Antoine[1,*], Isabelle Pison[1], Patrick M. Crill[2], Brett Thornton[2], Philippe Bousquet[1], Thibaud Thonat[1], Thomas Hocking[1], Joël Thanwerdas[1], Jean-Daniel Paris[1] and Marielle Saunois[1]

[1]Laboratoire des Sciences du Climat et de l'Environnement, CEA-CNRS-UVSQ, IPSL, Gif-sur-Yvette, France.
[2]Department of Geological Sciences, Stockholm University, SE-10691 Stockholm, Sweden.

**Correspondence:** A. Berchet (antoine.berchet@lsce.ipsl.fr)

**Abstract.**

Characterizing methane sources in the Arctic remains challenging, due to the remoteness, heterogeneity, and variety of such emissions. *In situ* campaigns provide valuable data sets to reduce these uncertainties. Here we analyse data from the summer 2014 SWERUS-C3 campaign in the eastern Arctic Ocean, offshore Siberia and Alaska. Total concentrations of methane, as well as relative concentrations of $^{12}CH_4$ and $^{13}CH_4$ were measured continuously during this campaign for 35 days in July and August. Using a chemistry-transport model, we link observed concentrations and isotopic ratios to regional emissions and hemispheric transport structures. A simple inversion system helped constrain source signatures from wetlands in Siberia and Alaska, and oceanic sources, as well as the isotopic composition of lower stratosphere air masses. The variation in the signature of low stratosphere air masses, due to strongly fractionating chemical reactions in the stratosphere, was suggested to explain a large share of the observed variability in isotopic ratios. These results point at necessary efforts to better simulate large scale transport and chemistry patterns to make a relevant use of isotopic data in remote areas. It is also found that constant and homogeneous source signatures for each type of emission in a given region (mostly wetlands, and oil and gas industry in our case at high latitudes) are not compatible with the strong synoptic isotopic signal observed in the Arctic. A regional gradient in source signatures is highlighted between Siberian and Alaskan wetlands, the latter having lighter signatures (more depleted in $^{13}C$). Finally, our results suggest that marine emissions of methane from Arctic continental shelf sources are dominated by thermogenic-origin methane, with a secondary biogenic source as well.

## 1 Introduction

Methane ($CH_4$ ) is both a potent greenhouse gas and a precursor of ozone with very diverse sources and sinks in the atmosphere (Saunois et al., 2016). The wide variety of $CH_4$ sources and their spatial and temporal heterogeneity make the uncertainties on $CH_4$ budgets very large, on both regional and global scales (Saunois et al., 2016). This impairs our understanding of the variations of atmospheric concentrations, particularly of which sources of methane and/or regions are causing these variations, which have been rapid in recent decades (Dlugokencky et al., 2009; Nisbet et al., 2016; Saunois et al., 2017; Nisbet et al., 2019; Turner et al., 2019).

In the Arctic, major $CH_4$ sources are natural wetlands, in-land waters (lakes, streams, deltas, estuaries), leaks from oil and gas extraction and transport, wildfires, seabed and geological seepage. The magnitude of all these sources suffers with very high uncertainties (McGuire et al., 2009; Kirschke et al., 2013; Berchet et al., 2015; Arora et al., 2015; Berchet et al., 2016; Ishizawa et al., 2019). The large areas of wetlands above 50°N and the high sensitivity of their $CH_4$ emissions to the changing climate make this zone a key region for the global $CH_4$ budget. The present uncertainties on $CH_4$ sources and sinks in the Arctic are very large, due to the complexity of the involved processes and the difficult access to these remote regions (e.g., Thornton et al., 2016b; Bohn et al., 2015). Moreover, in addition to increased $CH_4$ emissions from wetlands and thawing permafrost, increasing ocean temperatures could lead to the destabilization of methane hydrates on the Arctic continental shelf, potentially emitting large quantities of $CH_4$. For instance, significant point emissions have been detected along the East Siberian Arctic Shelf (Shakhova et al., 2010, 2014; Thornton et al., 2016a, 2020) taking the shape of $CH_4$ flaring from the sea floor extending up to the surface. However, upscaling point measurements of "hot spots" proves difficult and there is no proof that such methane hydrate emissions are currently reaching the atmosphere in large quantities (Berchet et al., 2016; Pisso et al., 2016; Ruppel and Kessler, 2017). Other potential Arctic seafloor sources of $CH_4$ include emissions from degrading subsea permafrost (Dmitrenko et al., 2011), leakage from natural gas reservoirs, and degrading terrestrial organic carbon transported onto the continental shelf (Charkin et al., 2011). $CH_4$ emissions from the Arctic would then have a positive feedback on climate change. A better knowledge of Arctic $CH_4$ emissions would reduce uncertainties in its global budget, and help to better quantify the sensitivity of Arctic regional sources and sinks to climate change.

For more than ten years, atmospheric measurements of methane concentrations have been performed in the Arctic, either at surface stations (e.g., Arshinov et al., 2009; Sasakawa et al., 2010; Dlugokencky et al., 2014), during mobile field campaigns such as the YAK-AEROSIB aircraft campaigns (Paris et al., 2010) and the TROICA train campaign (Tarasova et al., 2006, 2009) or during oceanographic campaigns (e.g., Pisso et al., 2016; Yu et al., 2015; Pankratova et al., 2019). In the present work, we analyze data from the SWERUS-C3 campaign on-board a ship in the Arctic Ocean during summer 2014 (Thornton et al., 2016a). Such short-term mobile campaigns are necessary to complement the limited number of long-term fixed, mostly coastal stations currently available. In particular, oceanic campaigns are expected to provide information on oceanic sources but also on land sources located upwind. However, $CH_4$ from various sources is being mixed during the atmospheric transport of the air masses, which makes it difficult to separate them without resorting to numerical modelling (Berchet et al., 2016).

Atmospheric inversions merge together observations, numerical modelling and emission data sets to attribute the observed variability in $CH_4$ concentrations to emitting regions and thus optimize the $CH_4$ budget. Such methods were successfully applied in the Arctic using *in situ* fixed stations (e.g., Berchet et al., 2015; Thompson et al., 2017; Ishizawa et al., 2019), as well as satellites when available (Tan et al., 2016). But despite technical progress in numerical modelling and inversion methods, it is hardly feasible to separate co-located emissions from different emitting sectors upwind observation sites based on observations of $CH_4$ concentrations alone. Observations of methane isotopic ratios could help separating emission sectors as the main emission processes are isotopically fractionating, causing significantly different isotopic source signatures. For example, high-latitude wetlands were attributed signatures in a range of $-80$ to $-55‰$ (Thornton et al., 2016b; Fisher et al., 2017; Ganesan et al., 2018). The $\delta^{13}$C-$CH_4$ signature of atmospheric $CH_4$ above the Arctic Ocean has been previously reported

in the range of $-50$ to $-47‰$ (Yu et al., 2015; Pankratova et al., 2019). Isotopes have already been used to characterise the origin of air masses in the Arctic (Fisher et al., 2011; Warwick et al., 2016), though these studies concluded that refinements in qualifying source emission isotopic signatures are required.

In the following, we explore the potential of using observations of isotopic ratios in the Arctic Ocean together with total CH$_4$ concentrations to separate pan-Arctic emission sources. We further analyse emission isotopic signatures in the Arctic from integrated atmospheric observations. We base our analysis on the unique observation set collected during the ship-based campaign SWERUS-C3 during summer 2014 in the Arctic Ocean. By comparing measurements to simulations of total CH$_4$ and isotopic ratio, we analyse to what extent the observable signal in the Arctic Ocean is exploitable in a numerical inversion system. In Sect. 2, we explain our inversion approach alongside giving details on the SWERUS-C3 observation campaign and on the model CHIMERE used in our study. In Sect. 3, we compare observations to simulations to assess the main contributions to the signal variability, and then implement a simplified inversion system to quantify isotopic emission signatures from various emission sectors around the Arctic.

## 2 Methods

### 2.1 Campaign and instrument description

Observations were carried out during the SWERUS-C3 campaign onboard the Swedish icebreaker *Oden* between July 14[th] and September 26[th], 2014. The cruise path was throught the central and outer Laptev and East Siberian seas, and finally the Chukchi Sea to Point Barrow, Alaska, in a first leg (see Fig. 1). A second leg of the cruise headed north from Point Barrow back through the Chukchi Sea and into the Arctic Ocean. As shown in Fig. S1 in supplementary materials, sea ice cover was present during a large portion of the campaign. Regions known to have active seafloor gas seeps (see Thornton et al., 2016a) occurred in both ice-free (in the Laptev Sea) and ice-covered (in the East Siberian Sea) regions.

Concentrations of total CH$_4$ were measured during the whole campaign using an off-axis cavity ring-down laser spectrometer, from Los Gatos Research (LGR) Inc. (Model 0010, FGGA 24EP, Mountain View, California, USA). Air inlets were located at 9, 15, 20, and 35 m above the sea surface; air was pulled through all inlets continuously, and analyzed from one inlet at a time for 2 minutes before switching to the next inlet. Data were filtered using wind speed and direction to avoid contamination from the ship exhaust. As no local sources influenced our measurements, concentrations are similar at all levels. We concatenate measurements from all inlets indifferently for our study. The spectrometer was calibrated every two hours using two synthetic air target gases; the target gases themselves were calibrated before, during, and after the cruise to two NOAA Earth System Research Laboratory certified standards for CH$_4$ . The reported precision was 0.5 ppb. Further details on the campaign conditions and instrument configuration are available in Thornton et al. (2016a).

Isotopic ratios were measured only during the first leg of the campaign, from July 14[th] to August 26[th] (see Fig. 1) using an Aerodyne Research, Inc (Billerica, MA, USA) direct absorption interband cascade laser spectrometer. This spectrometer measured the concentrations of the CH$_4$ isotopologues $^{12}$CH$_4$ , $^{13}$CH$_4$ , and CH$_3$D, the latter of which is not discussed in the current paper. The more common isotope ratio mass spectrometry methods directly provide (as their name implies) an

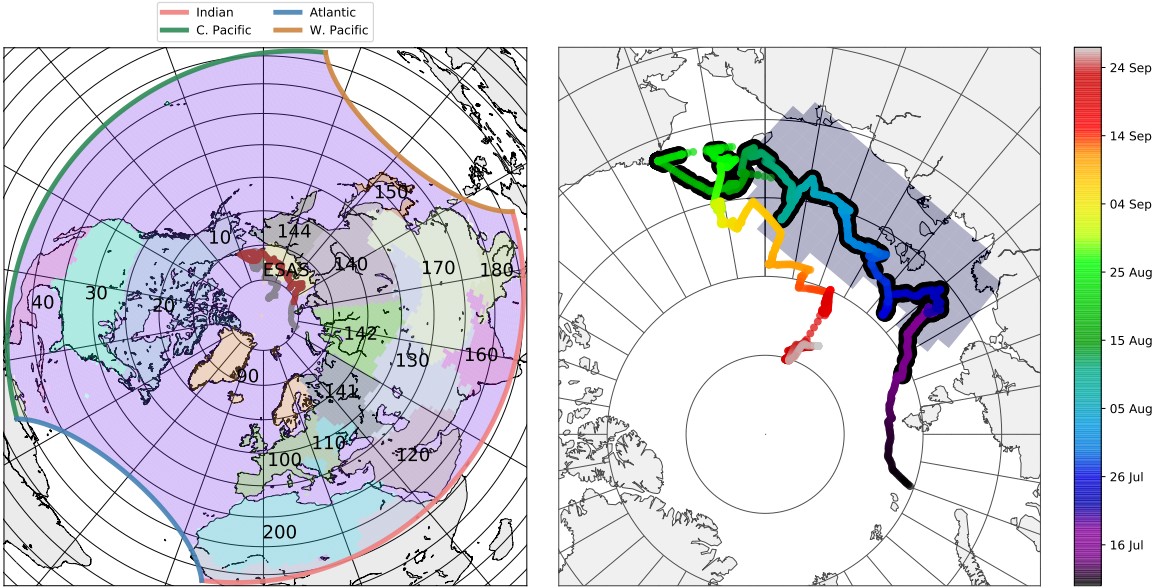

**Figure 1.** Path of the icebreaker *Oden* during the SWERUS-C3 campaign and domain of simulations. (left) The ship positions are represented by gray and brown dots, with brown points corresponding to locations where isotopic observations where carried out. The area delimited by coloured lines is the domain of CHIMERE simulations used for this study (see Sect. 2.2). The shaded areas and associated numbers correspond to the regions and their IDs used to separate contributions from remote emissions to the observed signal, as detailed in Sect. 2.3. ESAS = East Siberian Arctic Shelf. CHIMERE boundary conditions are split along the four sides of the domain as indicated by the coloured lines. (right) Zoom on the area covered by the campaign. The icebreaker's locations are coloured depending on their corresponding dates. Ship positions with a black edge are locations where isotopic observations where carried out. More details on the campaign in Thornton et al. (2016a). The shaded area corresponds to the ESAS emission region used in our simulation set up.

isotope ratio. In contrast, because the Aerodyne spectrometer measures the individual isotopologues, they must be individually calibrated before converting to $\delta^{13}$C-CH$_4$ values; this method is described in McCalley et al. (2014).

## 2.2 Model description

The Eulerian model CHIMERE (Menut et al., 2013) was run to simulate total concentrations of CH$_4$ as well as partial $^{12}$CH$_4$ and $^{13}$CH$_4$ concentrations to compute CH$_4$ isotopic ratios afterwards using the following formula:

$$\delta^{13}C = \frac{\left(\frac{[^{13}C]}{[^{12}C]}\right)_{sim}}{\left(\frac{[^{13}C]}{[^{12}C]}\right)_{ref}} - 1 \tag{1}$$

with $\left(\frac{[^{13}C]}{[^{12}C]}\right)_{ref} = 0.0112372$ the reference ratio from Craig (1957).

The domain of simulations spans over most of the Northern hemisphere with a horizontal resolution of $\sim 100\,\text{km}$ in order to include most contributions from distant sources (see Fig. 1). Similarly, the model uses 34 vertical levels from the surface up to $150\,\text{hPa}$ to represent stratosphere-to-troposphere intrusions. A spin-up period of six months prior to the campaign was used to properly assess the impact of air masses transported for long periods before reaching the Arctic ocean. The chemical sink of $CH_4$ by OH radicals is explicitly computed in CHIMERE using pre-computed fixed OH fields from the chemical model LMDZ-INCA (Hauglustaine et al., 2004; Folberth et al., 2006).

CHIMERE runs use the following input data streams: (i) meteorological fields downloaded from the European Centre for Medium Range Weather Forecasts (www.ecmwf.int) at $0.5°$ resolution every $3\,\text{hours}$; (ii) anthropogenic emissions aggregated at the CHIMERE resolution from the EDGARv4.3.2 database at $0.1°$ horizontal resolution (Crippa et al., 2016); (iii) wetland emissions interpolated from the model ORCHIDEE at $0.5°$ horizontal resolution (Ringeval et al., 2010); (iv) boundary $CH_4$ concentration fields extracted from the general circulation model LMDZ; these global simulations include both the chemical sinks of OH and chlorine, as well as their impact on the isotopic ratios; Cl and OH fields are prescribed offline from the chemical model LMDZ-INCA; (v) and isotopic signatures of the different sources chosen from Sherwood et al. (2017).

The chemical sink by chlorine is not included in our setup to keep simulations as light as possible. This sink can be separated into two main contributions: the upper stratosphere and the Arctic Ocean boundary layer. The upper stratosphere is not included in our model of simulation, but chlorine sink (and isotope fractionation) is expicitly accounted for in global LMDZ simulations used as boundary conditions in our setup. Regarding the Arctic Ocean boundary layer, the setup by Thonat et al. (2017) was adapted to our case, including the boundary layer Cl sink using pre-computed fields from the model LMDZ-INCA. It resulted in differences of concentrations lower than $1\,\text{ppb}$ over the Arctic ocean, and less than $0.02\text{‰}$ for the isotopic ratio of air masses, which is negligible compared to the signal we are inquiring into.

Other fluxes not included in our setup play a significant role in the regional pan-Arctic budget, such as in-land water bodies, wildfires and the sink in soil, but have limited impact on our observations. These fluxes were tested in our case and were quantified to cause differences in simulated concentrations lower than $2\,\text{ppb}$, and less than $0.01\text{‰}$ in simulated isotopic ratios at the locations sampled during the SWERUS-C3 campaign.

## 2.3 Atmospheric inversion of isotopic signature

Usually observations of $\delta^{13}$C-$CH_4$ are used to help constraining methane fluxes and differentiating between different sources with known signatures. However, the intrinsic spatial and temporal variability of source isotopic signatures limits the robustness of this approach (e.g., Fisher et al., 2017, as illustrated in Sect. 3.1). Here, we conversely assume that total $CH_4$ is properly simulated by our model (as confirmed by the good performance of the model to reproduce total $CH_4$ concentrations, highlighted in Sect. 3.1) and that the relative contributions of various sources from various regions are correct. Thus we use $\delta^{13}$C-$CH_4$ observations to help reduce uncertainties on source isotopic signatures: we test the ability of the ship-based measurements to help constrain the isotopic signature of remote sources, such as wetland sources and oceanic emissions from the Laptev, East Siberian, and Chukchi Seas, dominant in the region explored during the campaign.

To do so, $\delta^{13}$C-CH$_4$ observations are implemented into a classical analytical Bayesian framework (Tarantola, 2005). The designed inversion system optimizes source signatures from different source types and different regions. At every time step when an isotopic observation is available, the system fits observations of isotopic ratios by altering the isotopic ratio in air masses coming from relevant source types and regions. Thus, the control vector contains one isotopic ratio value to optimize for each time step, each sector, and each region as detailed in Eq. 3 below.

The isotopic ratios of wetlands, solid fossil fuels, oil and gas, other anthropogenic sources from various land regions, and a potential variety of marine sources (gas field leaks, decomposing hydrates, degrading permafrost, etc.) from the East Siberian Arctic Shelf (ESAS), as well as from air masses coming from the sides and roof of our domain of simulations are optimized in the system. Apart from ESAS, emissions are spatially differentiated into 23 geographical regions (see Figure 1). Contributions from different regions and sectors are differentiated by computing so-called response functions by region, emission type and boundary side. That is to say, we carry out individual CHIMERE chemistry-transport simulations for every region, every type of emission and every side of the domain, all the other emissions and boundary conditions being switched off, resulting in an ensemble of 98 response functions (= 23 regions $\times$ 4 sectors + ESAS + 4 sides + top).

The simulated isotopic final composition $\mathbf{y}(t)$ at every given time step $t$ when an observation is available is retrieved by scaling relative contributions according to assumed source signatures (or original average composition for boundary conditions) as follows:

$$\mathbf{y}(t) = \sum_{r\in\text{regions}} \sum_{s\in\text{sectors}} \alpha_{r,s}(t) \times \delta_{r,s}(t) \tag{2}$$

with $r$ and $s$ varying over all available regions and sectors respectively, $\alpha_{r,s}^t$ ($0 < \alpha_{r,s}^t < 1$) the relative contribution of the sector $s$ from region $r$ at time $t$ and $\delta_{r,s}^t$ the signature in ‰ of the sector $s$ from region $r$ at time $t$.

This linear relationship allows us to define the control vector $\mathbf{x}$ and the observation operator, linking the control vector to observations of isotopic ratios, to easily compute and scale the simulated isotopic composition:

$$\mathbf{y}(t) = \mathbf{H}(t)\mathbf{x}(t) \text{ with } \begin{cases} \mathbf{x}(t) &= \delta_{r,s}(t) \quad \forall (r,s) \in (\text{regions}) \times (\text{sectors}) \\ \mathbf{H}(t) &= (\alpha_{r,s}(t))_{r\in\text{regions},s\in\text{sectors}} \end{cases} \tag{3}$$

Given the prior control vector $\mathbf{x}^{\mathrm{b}}$ containing assumed source signatures before inversion, the observation vector $\mathbf{y}^{\mathrm{o}}$ and the observation operator $\mathbf{H}$, optimized signatures are obtained by solving the Bayesian problem equation:

$$\mathbf{x}^{\mathrm{a}} = \mathbf{x}^{\mathrm{b}} + \mathbf{K}(\mathbf{y}^{\mathrm{o}} - \mathbf{H}\mathbf{x}^{\mathrm{b}}) \tag{4}$$

with $\mathbf{K} = \mathbf{P}^{\mathrm{b}}\mathbf{H}^{\mathrm{T}}(\mathbf{R} + \mathbf{H}\mathbf{P}^{\mathrm{b}}\mathbf{H}^{\mathrm{T}})^{-1}$ the Kalman matrix.

The matrix $\mathbf{R}$ represents uncertainties in the observations and in the capability of the model to reproduce them. In our case, we set them uniformly to 1.5‰ (1‰ from observation errors and 0.5‰ from simulation errors). The matrix $\mathbf{P}^{\mathrm{b}}$ represents uncertainties and covariances in the prior knowledge we have on source signatures. We build the matrix $\mathbf{P}^{\mathrm{b}}$ following the

**Table 1.** Isotopic signatures for the inputs in the CHIMERE model. The min-max range is deduced from existing literature (Sherwood et al., 2017; Sapart et al., 2017). The prior signature is computed as the center of the min-max range.

| Emission type | Prior signature (‰) | Min-Max range (‰) | Temporal correlation scale (days) |
|---|---|---|---|
| Wetlands | -65 | 25 | 15 |
| Fossil solid | -55 | 25 | 30 |
| Oil & gas | -42 | 15 | 30 |
| Other anthropogenic | -60 | 10 | 30 |
| ESAS | -55 | 15 | 15 |
| Boundary concentrations (sides) | -47.5 | 0.5 | 7 |
| Boundary concentrations (top) | -47.5 | 1 | 7 |

values in Tab. 1, deduced from Sherwood et al. (2017) and Sapart et al. (2017). Ranges and prior signatures for boundary conditions are deduced from global simulations with the model LMDZ. Observation time steps are not optimized separately. Instead, we use temporal correlations in the $\mathbf{P}^b$ between different time steps. We represent temporal correlations between two time steps $t_i$ and $t_j$ as:

$$r = exp\left(-\frac{|t_i - t_j|}{\tau}\right) \tag{5}$$

with $\tau$ the temporal correlation scale of Tab. 1.

As shown in Tab. 1, the values of source signatures are not well known and a very large range of signatures is available in the literature. To account for this large variety of realistic signatures, we carry out a Monte Carlo ensemble of 8000 inversions with varying prior signatures and uncertainties, instead of running one single inversion. Prior signatures are sampled following a normal distribution with average and standard deviation from Tab. 1; the standard deviation is chosen as half of the min-max range. Uncertainties are sampled following a uniform distribution spanning over $[\sigma_{ref}/2, \sigma_{ref}]$, with $\sigma_{ref}$ equals half of the min-max range of Tab. 1.

In the end, we obtain hourly posterior signatures for each simulated sector and region for each of the 8000 inversions. Even though posterior signatures are available for each region and each sector at each observation time step, we do not inquire into the temporal variability of sources as constraints provided by the SWERUS observations are very heterogeneous in time and space. Instead, we compute overall posterior distributions for each simulated sector and region based on an ensemble of 8000000 (=8000 inversions × 1000 hourly observations). To minimizing the impact of control vector components that are ill-constrained by the inversion, all data points are not evenly counted in posterior distributions. Posterior distributions of signatures are computed accounting for all the Monte Carlo samples and weighted by the corresponding values of the sensitivity matrix $\mathbf{KH}$ (Cardinali et al., 2004), which gives an indicator of how much observations constrain one component of the control vector. The posterior optimal signature for each region and sector is computed as the maximum of the probability distribution.

# 3 Results and discussion

## 3.1 Forward modelling of total methane and isotopic ratio

Figure 2 shows observations of total $CH_4$ and of isotopic ratios as measured during the campaign and compared to simulations. The model CHIMERE reproduces well most of the variability in the total $CH_4$ signal. The average bias over the period is lower than 5 ppb with a correlation of 0.66 between observations and simulations on an hourly basis. Most peaks spanning more than one day are properly represented in the model, proving the capability of the model to reproduce the synoptic variability of the observations. Smaller peaks are missed by the model, in particular on Aug. 5, 12 and 15, indicating that some local sources are not included in the model, or are dispersed too quickly in the numerical realm. These could be local intense seeps met along the ship's track, or onshore wetlands not well represented with the model ORCHIDEE at $0.5°$ horizontal resolution. We do not investigate further missing emissions as most peaks are well explained by the model, which we assume sufficient to carry out an inversion of isotopic signatures as described in Sect. 3.2.

When computing the intersect with the y-axis of the linear fit between $\delta^{13}$C-$CH_4$ and total $CH_4$ (see Keeling plots in Supplementary material), the observed isotope ratios point to an average generic Arctic source of $-63.0‰$, consistent with dominant biogenic sources in Arctic regions. The model reproduces well this average signature at $-59.5‰$. Observations highlight a strong synoptic variability in isotopic ratios in the Arctic, with a standard deviation of $0.50‰$ and a range of $2‰$. Most of this is missed by the model (see Fig. 2, top panel, prior simulation). Simulated ratios with fixed (temporally and spatially) isotopic signatures for the emission sectors detailed in Sect. 2.3 barely exhibit any variations. The prior standard deviation is $0.22‰$ (resp. $0.12‰$ when removing the wetland event on August $21^{st}$), with a range of $1.5‰$ (resp. $0.5‰$). Considering the good fit of simulations to observations of total $CH_4$ , the missing variability indicates that the classical assumption of uniform signatures for given sectors and regions is not valid in the Arctic, consistent with Ganesan et al. (2018) and Fisher et al. (2011). Contributions to modelled concentrations from different regions of a given emission sector can change much more than the variability of total $CH_4$ as indicated in Fig. 2. For instance, on July $22^{nd}$, contributions from wetlands turn from a dominating Siberian influence to a North American one, causing a change of $\sim 30$ ppb in the signal. Differences in the average wetland source signatures between these two regions of $\sim 20‰$ (as suggested by Ganesan et al., 2018) would thus translate into $\sim 0.3‰$ in measured isotopic ratio, partly explaining the corresponding observed event (see middle panel of Fig. 2).

Still, more critical for the composition of air masses are the changes in very large-scale hemispheric contributions. As indicated by the blue shades in Fig. 2 (middle panel), depending on the dominant large scale transport patterns, contributions from the stratosphere and from the model lateral sides (located in the Tropics) can vary by more than 400 ppb within a few days. This corresponds to dominantly updraught or downdraught transport patterns, as illustrated by Fig. S2 in Supplement. These very strong variations in total $CH_4$ enhance the impact of uncertainties in the vertical and horizontal distribution of isotopic ratios at the hemispheric scale. First, tropical air masses are influenced by tropical wetlands and anthropogenic emissions, causing a spatial and temporal variability in tropical isotopic ratio of up to $1‰$, which is not accounted for in our CHIMERE setup with fixed isotopic ratios at the simulation domain sides (see Sect.2.2). Second, the vertical profiles of isotopic ratios in the Arctic (see simulated example from the global transport model LMDZ in Fig. S3 in Supplement) are very steep. Such

gradients are poorly represented in most global models, due to issues in the representation of the vertical transport or to the insufficiently quantified fractionating OH and chlorine sinks in the stratosphere and upper troposphere. These two sources of uncertainties in chemistry-transport models coupled with the strong real-world variations in stratospheric and tropospheric contributions could explain why the regional model CHIMERE does not reproduce the strong synoptic variability in $\delta^{13}$C-CH$_4$ observed during the SWERUS-C3 campaign. In particular, for the above-mentioned event of July 22$^{nd}$, contributions from the domain sides vary by more than 300 ppb. Such a variability in CH$_4$ contributions, associated with differences of a few ‰ between the isotopic ratios of lower stratosphere airmasses and mid/low latitude air masses, could explain the observed event.

Thus, the first order variability of atmospheric isotopic ratios is due to a balance between non-regional transport-related hemispheric features and regional contributions of wetland, ocean and anthropogenic emissions.

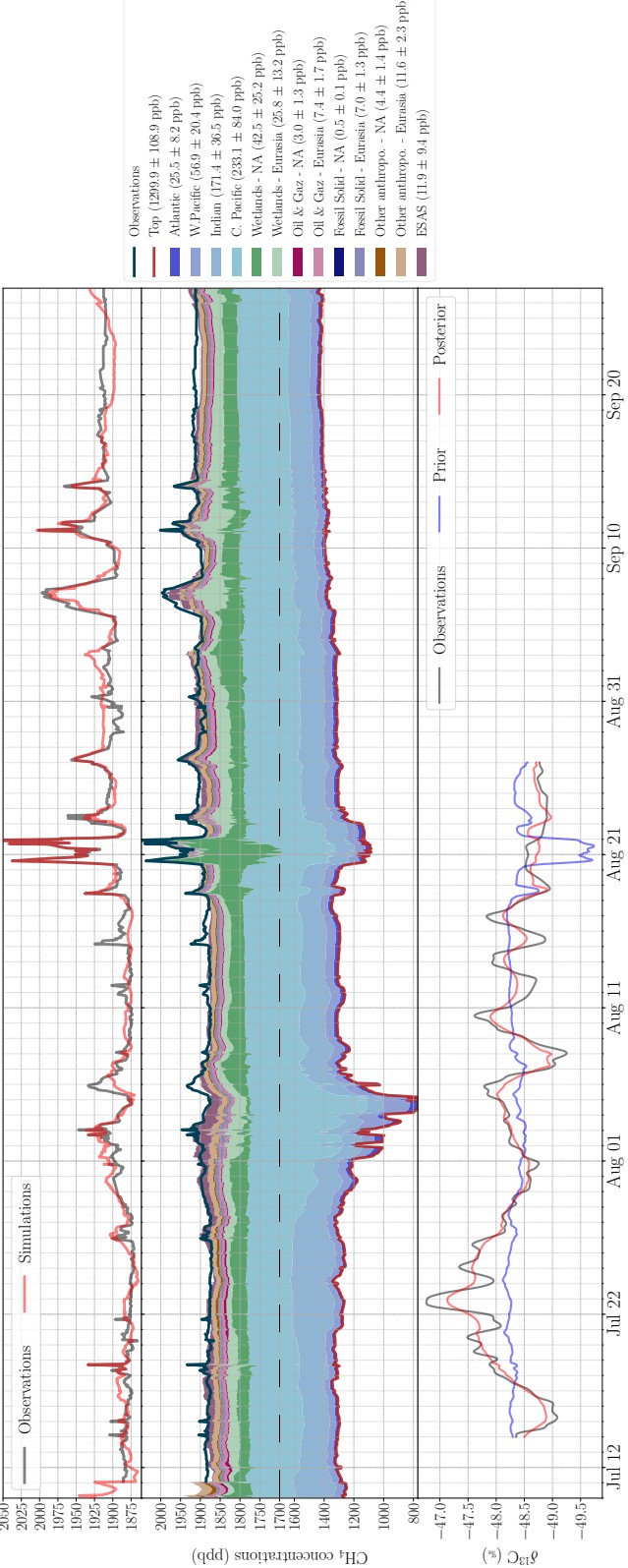

**Figure 2.** (top panel) Observed and simulated total CH$_4$ concentrations; (middle panel) simulated contributions to total CH$_4$ concentrations; individual regions simulated by the model (see Fig. 1) are aggregated into two main continental components: North-America (NA) and Eurasia; light green areas depict Eurasian (mostly Siberian) wetlands, while dark green ones are North American wetlands. Shaded blue areas represent contributions from the sides of the CHIMERE simulation domain (see Fig. 1); orange shades represent minor anthropogenic contribution. So-called "top" line gives the simulated concentrations originating from the lower stratosphere (i.e., from the top of CHIMERE simulation domain). Please note the gap in y-axis scale at 1700 ppb highlighted by the dash line. (bottom panel) Observed and simulated isotopic ratios before (prior) and after (posterior) inversion.

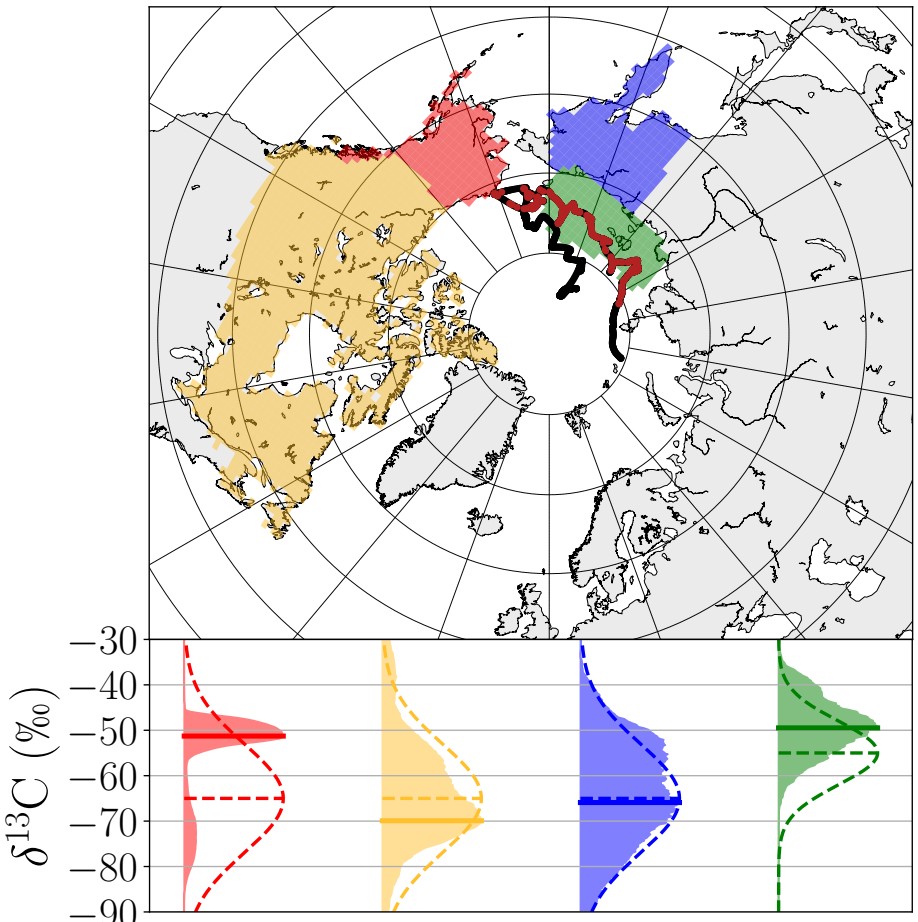

**Figure 3.** (top panel) Map of regions constrained by the observations in the inversion; for land regions, only wetlands are constrained; the green region corresponds to oceanic sources from the East Siberian Arctic Shelf; the ship path is indicated in black, with red points highlighting locations with available isotopes observations. (bottom panel) Distribution of posterior hourly signatures as deduced by the inversion for regions constrained by the observations for the ensemble of 8000 Monte-Carlo inversions (see details in Sect. 2.3. Prior signatures distributions (dashed lines) are those of Tab. 1. The optimal posterior signatures, defined as the maximum of the posterior distribution, is highlighted by plain horizontal lines.

## 3.2 Optimisation of Arctic source signatures

Assuming that the mix of $CH_4$ sources is correct, we now attempt to separate hemispheric and regional contributions by optimizing source signatures for a set of geographical regions and different emission sectors in the Arctic as detailed in Sect. 2.3. Posterior isotopic ratios in Fig. 2 (bottom panel) follow most of the variability in observations, indicating the inverse method does fit the observations in a satisfying way. The rest of the signal is within the observation uncertainties of 0.1‰. This proves

that even though the model is not perfect in representing the transport, it is reasonable to use simulated contributions to optimize isotopic signatures.

Figure 3 shows the posterior signature distributions as deduced from the 8000 Monte Carlo inversions for the four regions that are the most constrained by the observations, i.e. weighted by the sensitivity matrix as detailed in Sect. 2.3. Accounting for the sensitivity matrix, it appears that only the roof boundary conditions (i.e., air masses from the lower stratosphere), ESAS emissions (i.e., emissions from the Laptev, East Siberian, and Chukchi Seas) and wetland regions on the shores of the Arctic ocean are reasonably constrained by the SWERUS-C3 ship-based campaign. Even though anthropogenic emissions were optimized in our system, only the wetland emission sector is significantly constrained for land regions (Fig. 3). The lower stratosphere signatures span in a short range of $-48.5/-46.5‰$. Wetlands are suggested to have a heavier signature in Canada (optimal signature: $-69.9‰$) than in Eastern Siberia (optimal signature: $-65.9‰$, with a node of similar importance at $-55‰$), consistent with Ganesan et al. (2018) and the compilation by Thornton et al. (2016b). Wetlands in Alaska exhibit a narrow posterior distribution at $-51.3‰$, with a secondary mode at $-75‰$. Alaska is thus well constrained by the inversion. However, the final value may suggest that the inversion has difficulties in differentiating collocated emissions and mixes the signal due to thermogenic sources with co-located wetland emissions, as it is the case in Alaska with extensive extraction of raw oil and gas.

Posterior ESAS signatures are significantly shifted by more than $5‰$ to $-49.5‰$ from the prior signature towards lighter values. This compares with previous studies and points towards a mix of different processes taking place in the Arctic shelf such as inputs from the sea bed (James et al., 2016; Berchet et al., 2016; Skorokhod et al., 2016; Pankratova et al., 2018; Thornton et al., 2020). The posterior signature could thus be explained by mixed biogenic and thermogenic sources, confirming that ESAS emissions, possibly including an hydrate contribution, are not as depleted as wetland sources (Cramer et al., 1999; Lorenson, 1999).

Overall, the approach developed here reveals that the spatial and temporal variations of isotopic source signatures must be accounted for in order to properly represent $\delta^{13}$C-CH$_4$ observations. Such an approach does not allow us to reach definitive conclusions when considering the spread of the inferred regional isotopic signatures. However, it is crucial to account for isotopic ratios to avoid misallocating methane flux variations in methane inversions. We also show that atmospheric $\delta^{13}$C-CH$_4$ signals can be significant (larger than observation errors), indicating a good potential for the use of isotopic observations based on oceanic campaign to improve our knowledge of the Arctic methane cycle. Finally, the weight of the boundary conditions in the signal points at necessary progress in global simulations (including fractionating chemical reactions in the stratosphere) of CH$_4$ atmospheric isotopic ratios.

## 4 Conclusions

Observations of total atmospheric methane and isotopic ratio were carried out in Summer 2014 in the Arctic Ocean during the SWERUS-C3 campaign onboard the Swedish icebreaker *Oden*. A unique continuous dataset of 45 days of atmospheric isotopic ratios over the Arctic Ocean is available from this campaign. Consistently with other campaigns in the region collecting flasks,

the synoptic variability of atmospheric isotopic ratios in the Arctic is very strong, spanning $\sim 2‰$, largely above observation error. Using forward simulations, we confirmed that the assumption of uniform isotopic signatures to represent emission sectors is invalid in the Arctic dominated by natural sources. We also exhibited the strong dependency of atmospheric isotopic ratios to large-scale changes in air mass origin (lateral boundaries of our simulation domain, corresponding to mid-/low-latitude air masses; top boundaries corresponding to lower stratosphere air masses). Based on a simplified inversion framework, the SWERUS-C3 data were used to infer isotopic source signatures of the Arctic regions and emission sectors. Due to the limited number of available observations and the important distance between sources and observations, our system was not able to provide any significant constraints on anthropogenic emissions, and could optimize signatures from ESAS and wetlands near the Arctic Ocean shores only. Wetland and oceanic ESAS source signatures were found to span a very wide range with a multimodal distribution for wetlands. The inversion also indicated that $CH_4$ emissions from ESAS are composed of a mixture of dominant thermogenic methane, complemented by some biogenic methane.

Overall, only a strong spatial and temporal variability in emission signatures and in stratospheric isotopic ratios can explain the variability of observations. Therefore, our study points at necessary improvements in simulating the first-order transport and chemistry of methane and its isotopes to reproduce large scale hemispheric features, especially stratosphere to troposphere exchanges. This makes it necessary to improve i) the quality of continuous isotopic measurements to capture the synoptic signal with even higher confidence, ii) numerical chemistry-transport models, so that the uncertainties on the first-order processes are at least one order of magnitude smaller than the regional signal, which is not the case in our study, and iii) the mapping of isotopic emission signatures used as priors in inversions as initiated by Ganesan et al. (2018).

**Author contributions**

AB, TT and IP designed the simulation experiments. AB and TT developed the code and performed the CHIMERE numerical simulations. TH and JT ran global simulations. PC and BT designed, carried out and provided observation data from the SWERUS-C3 campaign. PB, MS and JDP contributed to the scientific analysis of this work. AB prepared the manuscript with contributions from co-authors.

**Code/Data availability**

The isotopic ratio and total $CH_4$ observation data are available upon request. The transport model CHIMERE can be downloaded here: https://www.lmd.polytechnique.fr/chimere/. CHIMERE simulations were driven by the inversion system PYVAR-CHIMERE (Fortems-Cheiney et al., 2019). Inputs for our CHIMERE simulations are freely available following instructions detailed in respective references in Sect. 2.2.

**Competing interests**

The authors declare no competing interests.

*Acknowledgements.* We thank the crew of I/B *Oden* who made the SWERUS-C3 expedition possible. SWERUS-C3 funding was provided by the Knut and Alice Wallenberg Foundation, Vetenskapsrådet (Swedish Research Council), Stockholm University, Swedish Polar Research Secretariat, and the Bolin Centre for Climate Research. $\delta^{13}$C-CH$_4$ observations from SWERUS-C3 campaign will be posted to the Bolin Centre database, https://bolin.su.se/data/. This work has been supported by the Swedish Research Council VR through a Franco-Swedish

5     project named IZOMET-FS "Distinguishing Arctic CH$_4$ sources to the atmosphere using inverse analysis of high-frequency CH$_4$ , $\delta^{13}$C-CH$_4$ and CH$_3$D measurements" (grant no. VR 2014-6584). The study extensively relies on the meteorological data provided by the ECMWF. Calculations were performed using the computing resources of LSCE, maintained by François Marabelle and the LSCE IT team.

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
