# Peer review of "Using ship-borne observations of methane isotopic ratio in the Arctic Ocean to understand methane sources in the Arctic"

_Atmospheric Chemistry and Physics, 2019_

## Referee Comment (RC1) · Anonymous Referee #1 · 27 Oct 2019

The presented manuscript describes an attempt to improve the characterization of the spatio-temporal variability of methane isotope ratios from various sources in the high Northern latitudes. The analyses are based on ship-borne measurements of total CH4 and isotope ratios during summer 2014, which are interpreted with a chemistry transport model and simple atmospheric inversions. As opposed to using isotope ratios to better constrain regional methane emissions and their source types, the authors test herein whether or not the assumption of homogeneous and constant methane isotope ratios for specific source types is plausible. Their findings indicate that there are indeed regional scale gradients in isotope ratios from wetland emissions, and also sources from the East Siberian Shelf appear to be heterogeneous.

The authors postulate that methane emissions from the high northern latitudes are a highly important component of the global greenhouse gas budget, and that they are likely to become more important with Arctic warming in the future. At the same time, the observational infrastructure in the region is very limited, so that isotope observations could become a valuable addition to existing data in order to better constrain the types of methane emission sources, and their location of origin. I fully agree with these statements, therefore I consider their presented attempt to better constrain the spatio-temporal variability in methane source isotope ratio signatures as very important to the research field. Unfortunately, there are some shortcomings in the presentation of the methodology. Also, sensitivity studies on some components of the study setup should be added to facilitate an evaluation of the quantitative results. Detail on these major comments are listed below:

1.) As much as I appreciate a concisely written paper, in this case the description of the methodology ended up being too short, so some characteristics of the optimization setup remain unclear. What exactly is optimized in this approach? In the setup, there are four different source types given for terrestrial areas, plus ESAS and boundary conditions (Table 1), and there are 24 regions differentiated (Figure 1). I assume that the authors optimized only wetland emissions from each terrestrial region, plus ESAS emissions and the 5 boundary conditions, which would make 30 free parameters to constrain (they discuss only 3 optimized parameters/source distributions in the results section ..). However, this is nowhere clearly documented, and given the available combinations of source types and regions, the total number could as well be 102. Also, the source of the starting values for the isotope ratio is given, but not the source of the uncertainty ranges. It may be the same, but this should be documented more clearly.

2.) This paper presents some evidence that the source signatures from wetlands is deviating between different Arctic terrestrial regions, but the accuracy of the results is not tested adequately. The authors provide a short statement that, given the good agreement between observations and simulations of total methane, they assume that

the transport model is correct here (Section 2.3) Also, they provide fixed uncertainty ranges for prior isotope ratios, but fail to clearly document where these are coming from, and how trustworthy they are .The latter may be an easy fix, while the former is clearly reducing the impact of these results. I therefore request to add sensitivity studies where the role of transport and prior uncertainties on the outcome of the study is tested and quantified. It shouldn't be too much extra work to perturb the transport, and check if this results in major shifts in optimized isotope ratios for different regions. The same holds true for the assignment of prior uncertainties.

Overall, this short and (in part too) concisely written paper highlights the need for a better characterization of isotope ratios in different source types and source regions for methane around the Arctic. The overall message, i.e. that there is a large potential in isotope ratio observations to help constrain methane sources, but we need to do a better job in characterizing them, is both important in clearly articulated here. The use of figures and tables is adequate. I therefore recommend the publication of this manuscript in Atmospheric Chemistry and Physics, once the major and minor (see below) comments I raised herein have been addressed.

ADDITIONAL COMMENTS: The introduction could be strengthened by additional material on existing attempts to constrain Arctic CH4 sources in different regions, with different approaches. Where does the controversy on ESAS emissions stem from, and where are the knowledge gaps? How exactly will this campaign contribute to filling these gaps?

As mentioned above, the Methods section is very short, sometimes too short. For the interested reader more details on the methodology (e.g. on the CHIMERE model) should be provided either in an appendix, or in an SI section. In Section 3.3, more details on the optimization target and strategy need to be provided also in the main text (see details above).

Figure 2 raises the question why the cumulative contributions of top and 4x side com-

[Figure]

ponents are so stable over time, given the pronounced variability in contributions from different sides of the model domain, which is also discussed in Section 3.1? Maybe it would be helpful to show the mean concentrations from the global model that were taken as a reference for the 5 boundary values applied here.

One confusing part of Section 3.2 is the mentioning of 'temporal variability' - what are the authors referring to? Obviously, there was no temporal variability in the source isotope ratios detected, or at least none is shown (or discussed) in any of the results. The only temporal variability shown is that of the retrieved isotope signal in the SWERUS data (Figure 2, bottom), but here the variability has been largely attributed to shifts in the origin of air masses, and accordingly source regions. The statement on temporal variability is repeated also in the conclusions. Please clarify.

Finally, please make sure your reference list is up-to-date (old discussion papers cited).

---

## Referee Comment (RC2) · Anonymous Referee #2 · 5 Nov 2019

Understanding Arctic methane is of great importance because it is region where there is a large volume of frozen carbon that is highly susceptible to global warming. There are few continuous monitoring stations that monitor changes in Arctic methane emissions, fewer still measuring methane isotope ratios. This paper describes an Arctic ship-borne measurement campaign that measures total CH4 and isotope ratios. These data are used in an atmospheric transport model and a simple inversion system to characterise the variability of methane source signatures from Arctic source types. Usually, isotope data are used in an inversion to improve the attribution of emission estimates, an approach that assumes the isotopic source signatures are known. Here, the authors assume the methane emissions are known and instead solve for isotopic source

signatures from the Arctic region. They find there is variability in the source signatures and therefore it is not appropriate to assume constant emissions source signatures for Arctic methane. The work described in the manuscript will make a useful contribution to the literature by further understanding methane isotopic ratios and their importance to climate change in the Arctic. This paper is within the scope of ACP and presents a novel dataset in an interesting way. I support the work being published in ACP after the authors have addressed my comments and queries below, which I believe will make their work clearer to understand.

In general, I found the methodology could be clearer and some of the decisions in the data analysis approach could be justified (in some cases) and justified more thoroughly (in other cases). Without such additional information I would find it difficult to reproduce their results given their dataset.

In particular, in section 2.3, I would like to see the inversion approach described in more detail. It should be made completely clear what they are solving for in their state vector and which parameters go into the inversion. The approach of the inversion is also unclear to this reviewer. This could be rectified by detailing the inversion method, and then laying out the relevant equations (perhaps in the supplementary material). In section 2.2, it sounds as though only OH is included as a sink, which ignores other potentially important sinks (such as the soil sink, Cl radical and stratospheric loss). Similarly, these sinks seemingly aren't accounted for in the optimisation, despite the fact that they also have larger uncertainties that vary in space and time. I am not suggesting the authors solve for them but they should at least acknowledge these other sinks for completeness. In section 2.2, it seems that only wetlands and anthropogenic emissions are used in the model, but again that is not clear from reading the manuscript. I suggest to address this explicitly in the main text or summarize the information in a table that lays out all the sources used in the model.

Building on this, it would be useful if the decisions that went into the setup were justified more clearly. For example, the authors describe that they assume emissions and

atmospheric transport from the model are suitable due to how they match observations, but this is not rigorously tested. It might be possible to first solve for emissions and then for source signature (making a two-step inversion), or to perform sensitivity tests on their assumptions by, for example, perturbing model transport. Similarly, the lack of sinks other than OH, and the choice of which sources that are included, are not justified. The latter should be quite simple to rectify with a short explanation and some literature examples, or by laying out a table of the current understanding of methane source signatures.

Some minor corrections include spelling and grammar errors. Some sentences do not make sense, e.g. the first sentence in the abstract. In addition, some further details of the how the model works could go into the supplementary information section. Figure 2 could be made clearer – it's a little hard to discern the observation line from the model results in the top panel. Maybe a separate panel with total model and observations would solve this. Also, the introduction would be stronger if it laid out current understanding of Arctic methane source types, thereby justifying the article by highlighting our current lack of understanding.

---

## Author Comment (AC1) · 7 Feb 2020

**Final response for "Using ship-borne observations of methane isotopic ratio in the Arctic Ocean to understand methane sources in the Arctic"**

Berchet Antoine[1,*], Isabelle Pison[1], Patrick M. Crill[2], Brett Thornton[2], Philippe Bousquet[1], Thibaud Thonat[1], Thomas Hocking[1], Joël Thanwerdas[1], Jean-Daniel Paris[1] and Marielle Saunois[1]

[1]Laboratoire des Sciences du Climat et de l'Environnement, CEA-CNRS-UVSQ, IPSL, Gif-sur-Yvette, France.
[2]Department of Geological Sciences, Stockholm University, SE-10691 Stockholm, Sweden.

**Correspondence:** A. Berchet (antoine.berchet@lsce.ipsl.fr)

**1  Introductory replies**

We thank the referees for their time and for giving fruitful comments and reviews to our manuscript. It helped improving our manuscript substantially. We address below their comments and implement corresponding corrections to the manuscript.

Comments from referee #1 and #2 are reported below in blue and red respectively. We include point-by-point replies and
5  corresponding corrections to the manuscript are included between horizontal lines.

**2  General comments**

**2.1  Method section and material description**

1. **As much as I appreciate a concisely written paper, in this case the description of the methodology ended up being too short, so some characteristics of the optimization setup remain unclear. What exactly is optimized in this**
10  **approach? In the setup, there are four different source types given for terrestrial areas, plus ESAS and boundary conditions (Table 1), and there are 24 regions differentiated (Figure 1). I assume that the authors optimized only wetland emissions from each terrestrial region, plus ESAS emissions and the 5 boundary conditions, which would make 30 free parameters to constrain (they discuss only 3 optimized parameters/source distributions in the results section ..). However, this is nowhere clearly documented, and given the available combinations of source types and**
15  **regions, the total number could as well be 102. Also,the source of the starting values for the isotope ratio is given, but not the source of the uncertainty ranges. It may be the same, but this should be documented more clearly.**

   **In general, I found the methodology could be clearer and some of the decisions in the data analysis approach could be justified (in some cases) and justified more thoroughly (in other cases). Without such additional information I would find it difficult to reproduce their results given their dataset.**

**In particular, in section 2.3, I would like to see the inversion approach described in more detail. It should be made completely clear what they are solving for in their state vector and which parameters go into the inversion. The approach of the inversion is also unclear to this reviewer. This could be rectified by detailing the inversion method, and then laying out the relevant equations (perhaps in the supplementary material).**

We agree that the method section was detailed enough. We fully rewrote the corresponding section to allow the reader to fully understand what is done in our system. In particular, the equations linking the relative contributions of simulated sectors and regions and their isotopic signatures to the simulated isotopic ratio are given, as well as those describing the control vector and the inversion problem.

Below is the amended section:
* * *
To do so, $\delta^{13}$C-CH$_4$ observations are implemented into a classical analytical Bayesian framework (Tarantola, 2005). The designed inversion system optimizes source signatures from different source types and different regions. At every time step when an isotopic observation is available, the system fits observations of isotopic ratios by altering the isotopic ratio in air masses coming from relevant source types and regions. Thus, the control vector contains one isotopic ratio value to optimize for each time step, each sector, and each region as detailed in Eq. 2 below.

The isotopic ratios of wetlands, solid fossil fuels, oil and gas, other anthropogenic sources from various land regions, and a potential variety of marine sources (gas field leaks, decomposing hydrates, degrading permafrost, etc.) from the East Siberian Arctic Shelf (ESAS), as well as from air masses coming from the sides and roof of our domain of simulations are optimized in the system. Apart from ESAS, emissions are spatially differentiated into 23 geographical regions (see Figure 1). Contributions from different regions and sectors are differentiated by computing so-called response functions by region, emission type and boundary side. That is to say, we carry out individual CHIMERE chemistry-transport simulations for every region, every type of emission and every side of the domain, all the other emissions and boundary conditions being switched off, resulting in an ensemble of 98 response functions (= 23 regions $\times$ 4 sectors + ESAS + 4 sides + top).

The simulated isotopic final composition $\mathbf{y}(t)$ at every given time step $t$ when an observation is available is retrieved by scaling relative contributions according to assumed source signatures (or original average composition for boundary conditions) as follows:

$$\mathbf{y}(t) = \sum_{r \in \text{regions}} \sum_{s \in \text{sectors}} \alpha_{r,s}(t) \times \delta_{r,s}(t) \tag{1}$$

with $r$ and $s$ varying over all available regions and sectors respectively, $\alpha_{r,s}^t$ ($0 < \alpha_{r,s}^t < 1$) the relative contribution of the sector $s$ from region $r$ at time $t$ and $\delta_{r,s}^t$ the signature in ‰ of the sector $s$ from region $r$ at time $t$.

This linear relationship allows us to define the control vector $\mathbf{x}$ and the observation operator, linking the control vector to observations of isotopic ratios, to easily compute and scale the simulated isotopic composition:

$$\mathbf{y}(t) = \mathbf{H}(t)\mathbf{x}(t) \text{ with } \begin{cases} \mathbf{x}(t) & = & \delta_{r,s}(t) \quad \forall (r,s) \in (\text{regions}) \times (\text{sectors}) \\ \mathbf{H}(t) & = & (\alpha_{r,s}(t))_{r \in \text{regions}, s \in \text{sectors}} \end{cases} \tag{2}$$

Given the prior control vector $\mathbf{x}^{\mathrm{b}}$ containing assumed source signatures before inversion, the observation vector $\mathbf{y}^{\mathrm{o}}$ and the observation operator $\mathbf{H}$, optimized signatures are obtained by solving the Bayesian problem equation:

$$\mathbf{x}^{\mathrm{a}} \quad = \quad \mathbf{x}^{\mathrm{b}} + \mathbf{K}(\mathbf{y}^{\mathrm{o}} - \mathbf{H}\mathbf{x}^{\mathrm{b}}) \tag{3}$$

with $\mathbf{K} = \mathbf{P}^{\mathrm{b}}\mathbf{H}^{\mathrm{T}}(\mathbf{R} + \mathbf{H}\mathbf{P}^{\mathrm{b}}\mathbf{H}^{\mathrm{T}})^{-1}$ the Kalman matrix.

The matrix $\mathbf{R}$ represents uncertainties in the observations and in the capability of the model to reproduce them. In our case, we set them uniformly to $1.5‰$ ($1‰$ from observation errors and $0.5‰$ from simulation errors). The matrix $\mathbf{P}^{\mathrm{b}}$ represents uncertainties and covariances in the prior knowledge we have on source signatures. We build the matrix $\mathbf{P}^{\mathrm{b}}$ following the values in Tab. 1, deduced from Sherwood et al. (2017) and Sapart et al. (2017). Ranges and prior signatures for boundary conditions are deduced from global simulations with the model LMDZ. The temporal correlation is used to represent exponentially decreasing temporal correlation between the signature errors at two given times for the same sector and region.

As shown in Tab. 1, the values of source signatures are not well known and a very large range of signatures is available in the literature. To account for this large variety of realistic signatures, we carry out a Monte Carlo ensemble of 8000 inversions with varying prior signatures and uncertainties, instead of running one single inversion. Prior signatures are sample following a normal distribution with average and standard deviation from Tab. 1; the standard deviation is chosen as half of the min-max range. Uncertainties are sampled following a uniform distribution spanning over $[\sigma_{\mathrm{ref}}/2, \sigma_{\mathrm{ref}}]$, with $\sigma_{\mathrm{ref}}$ equals half of the min-max range of Tab. 1.
* * *
2. **In section 2.2, it sounds as though only OH is included as a sink, which ignores other potentially important sinks (such as the soil sink, Cl radical and stratospheric loss). Similarly, these sinks seemingly aren't accounted for in the optimisation, despite the fact that they also have larger uncertainties that vary in space and time. I am not suggesting the authors solve for them but they should at least acknowledge these other sinks for completeness. In section 2.2, it seems that only wetlands and anthropogenic emissions are used in the model, but again that is not clear from reading the manuscript. I suggest to address this explicitly in the main text or summarize the information in a table that lays out all the sources used in the model.**

We agree that our framework was not fully comprehensive in terms of both emissions and sinks processes. However, in order to keep the inversion framework simple, we did not include sources/sinks with negligible influence. We controled

that assumption using corresponding simulations. In terms of emissions, wild fires and in-land water systems play a significant role at the pan-Arctic scale, but once mixed and transported over long distances, corresponding simulated contributions are negligible (less than 2 ppb) in the region sampled during the SWERUS campaign. Similarly, the terrestrial soil sink is significant at the regional scale, but induce perturbations of less than 1 ppb at the sampling locations. The impact on isotopic ratios is below 0.02‰ for all these processes.

Regarding the chlorine sink, we acknowledge that the very strong fractionation of the reaction with $CH_4$ may have a significant impact on concentrations and isotopic ratios. This sink occurs mainly in the upper stratosphere and in the Arctic Ocean boundary layer, The reaction in the stratosphere is included in the driving LMDZ simulations at the boundary conditions, and in the stratospheric part inside the domain, the residence time of air masses is too short to enable a significant impact. Regarding the chlorine sink in the boundary layer, when including the sink in our simulations, we saw an impact below 1 ppb and smaller that 0.01‰ in the isotopic ratios.

We clarify all these aspects in the updated version of the manuscript:
* * *
CHIMERE runs use the following input data streams: (i) meteorological fields were downloaded from the European Centre for Medium Range Weather Forecasts (www.ecmwf.int) at 0.5° resolution every 3 hours; (ii) anthropogenic emissions were aggregated at the CHIMERE resolution from the EDGARv4.3.2 database at 0.1° horizontal resolution (Crippa et al., 2016); (iii) wetland emissions were interpolated from the model ORCHIDEE at 0.5° horizontal resolution (Ringeval et al., 2010); (iv) boundary $CH_4$concentration fields were extracted from the general circulation model LMDZ; these global simulations include both the chemical sinks of OH and chlorine, as well as their impact on the isotopic ratios; (v) and isotopic signatures of the different sources were chosen from Sherwood et al. (2017).

The chemical sink by chlorine is not included in our set-up to keep simulations as light as possible. This sink can be separated into two main contributions: the upper stratosphere and the Arctic Ocean boundary layer. The upper stratosphere is not included in our model of simulation, but chlorine sink (and isotope fractionation) is expicitly accounted for in global LMDZ simulations used as boundary conditions for CHIMERE. Regarding the Arctic Ocean boundary layer, the set-up by Thonat et al. (2017) was adapted to our case, including boundary layer Cl sink using pre-computed fields from the model LMDZ-INCA. It resulted in differences of concentrations lower than 1 ppb over the Arctic ocean, and less than 0.02‰ for the isotopic ratio of air masses, which is negligible compared to the signal we are inquiring into.

Other fluxes not included in our set-up play a significant role in the regional pan-Arctic budget, such as in-land water bodies, wild fires and the sink in soil, but have limited impact on our observations. These fluxes were tested in our case and were quantified to cause differences in simulated concentrations lower than 2 ppb, and less than 0.01‰ in simulated isotopic ratios at the locations sampled during the SWERUS-C3 campaign.
* * *
**2.2   Uncertainties and sensitivity**

**Building on this, it would be useful if the decisions that went into the setup were justified more clearly. For example, the authors describe that they assume emissions and atmospheric transport from the model are suitable due to how they match observations, but this is not rigorously tested. It might be possible to first solve for emissions and then for source signature (making a two-step inversion), or to perform sensitivity tests on their assumptions by, for example, perturbing model transport. Similarly, the lack of sinks other than OH, and the choice of which sources that are included, are not justified. The latter should be quite simple to rectify with a short explanation and some literature examples, or by laying out a table of the current understanding of methane source signatures.**

**This paper presents some evidence that the source signatures from wetlands is deviating between different Arctic terrestrial regions, but the accuracy of the results is not tested adequately. The authors provide a short statement that, given the good agreement between observations and simulations of total methane, they assume that the transport model is correct here (Section 2.3). Also, they provide fixed uncertainty ranges for prior isotope ratios, but fail to clearly document where these are coming from, and how trustworthy they are. The latter may be an easy fix, while the former is clearly reducing the impact of these results. I therefore request to add sensitivity studies where the role of transport and prior uncertainties on the outcome of the study is tested and quantified. It shouldn't be too much extra work to perturb the transport, and check if this results in major shifts in optimized isotope ratios for different regions. The same holds true for the assignment of prior uncertainties.**

We thank the reviewers for pointing at this caveats in our work. We now give more details about the choices of the set-up in the Method section as explained above.

Regarding the uncertainties on our results, the original manuscript was indeed lacking a proper quantification of the robustness of our system outputs. We completed our work by doing a large ensemble of 8000 Monte-Carlo inversions with varying prior isotopic signatures and uncertainties of signatures. This allows us to include uncertainties properly. We do not explicitly include uncertainties in the transport, but they are included implicitly thanks to the $\mathbf{R}$ matrix, or observational matrix, which includes both instrument and transport errors. The posterior results are well within the instrument error, suggesting that further investigation on the model side are not necessary at this point (they may be necessary with new improved data sets with lower instrumental uncertainties).

Altogether, the ensemble of inversions points to similar conclusions, with a robust inclusion of uncertainties. Figure 3 and 4 were merged together for better clarity of the results.

Section 3.2 of the results was modified as follows:
* * *
Assuming that the mix of $CH_4$ sources is correct, we now attempt to separate hemispheric and regional contributions by optimizing source signatures for a set of geographical regions and different emission sectors in the Arctic as detailed in Sect. 2.3. Posterior isotopic ratios in Fig. 2 (bottom panel) follow most of the variability in observations, indicating the inverse method does fit the observations in a satisfying way. The rest of the signal is within the observation uncertainties of $0.1‰$. This proves

that even though the model is not perfect in representing the transport, it is reasonable to use simulated contributions to optimize isotopic signatures.

Figure 3 shows the posterior signature distributions as deduced from the 8000 Monte Carlo inversions for the four regions that are the most constrained by the observations, i.e. weighted by the sensitivity matrix as detailed in Sect. 2.3. Accounting for the sensitivity matrix, it appears that only the roof boundary conditions (i.e., air masses from the lower stratosphere), ESAS emissions (i.e., emissions from the Laptev, East Siberian, and Chukchi Seas) and wetland regions on the shores of the Arctic ocean are reasonably constrained by the SWERUS-C3 ship-based campaign. Even though anthropogenic emissions were optimized in our system, only the wetland emission sector is significantly constrained for land regions (Fig. 3). The lower stratosphere signatures span in a short range of $-48.5/-46.5‰$. Wetlands are suggested to have a heavier signature in Canada (optimal signature: $-69.9‰$) than in Eastern Siberia (optimal signature: $-65.9‰$, with a node of similar importance at $-55‰$), consistent with Ganesan et al. (2018) and the compilation by Thornton et al. (2016). Wetlands in Alaska exhibit a narrow posterior distribution at $-51.3‰$, with a secondary mode at $-75‰$. Alaska is thus well constrained by the inversion. However, the final value may suggest that the inversion has difficulties in differentiating collocated emissions and mixes the signal due to thermogenic sources with co-located wetland emissions, as it is the case in Alaska with extensive extraction of raw oil and gas.

Posterior ESAS signatures are significantly shifted by more than $5‰$ to $-49.5‰$ from the prior signature towards lighter values. This compares with previous studies and points towards a mix of different processes taking place in the Arctic shelf such as inputs from the sea bed (James et al., 2016; Berchet et al., 2016; Skorokhod et al., 2016; Pankratova et al., 2018). The posterior signature could thus be explained by mixed biogenic and thermogenic sources, confirming that ESAS emissions, possibly including an hydrate contribution, are not as depleted as wetland sources (Cramer et al., 1999; Lorenson, 1999).

Overall, the approach developed here reveals that the spatial and temporal variations of isotopic source signatures must be accounted for in order to properly represent $\delta^{13}$C-CH$_4$ observations. Such an approach does not allow us to reach definitive conclusions when considering the spread of the inferred regional isotopic signatures. However, it is crucial to account for isotopic ratios to avoid misallocating methane flux variations in methane inversions. We also show that atmospheric $\delta^{13}$C-CH$_4$ signals can be significant (larger than observation errors), indicating a good potential for the use of isotopic observations based on oceanic campaign to improve our knowledge of the Arctic methane cycle. Finally, the weight of the boundary conditions in the signal points at necessary progress in global simulations (including fractionating chemical reactions in the stratosphere) of CH$_4$ atmospheric isotopic ratios.

**3  Technical comments**

1. **The introduction could be strengthened by additional material on existing attempts to constrain Arctic CH4 sources in different regions, with different approaches. Where does the controversy on ESAS emissions stem**

**from, and where are the knowledge gaps? How exactly will this campaign contribute to filling these gaps?**

We added references and explanation about the ESAS controversy in the introduction following the referee's recommendation
* * *
Moreover, in addition to increased $CH_4$ emissions from wetlands and thawing permafrost, increasing ocean temperatures could lead to the destabilization of methane clathrates on the Arctic continental shelf, potentially emitting large quantities of $CH_4$. For instance, significant point emissions have been detected along the East Siberian Arctic Shelf (Shakhova et al., 2010) taking the shape of $CH_4$ flaring from the sea floor extending up to the surface. However, upscaling point measurements of hot spots proves difficult and there is no proof that such methane hydrate emissions are currently reaching the atmosphere in large quantities (Berchet et al., 2016; Pisso et al., 2016; Ruppel and Kessler, 2017). Other potential Arctic seafloor sources of $CH_4$ include emissions from degrading subsea permafrost (Dmitrenko et al., 2011), leakage from natural gas reservoirs, and degrading terrestrial organic carbon transported onto the continental shelf (Charkin et al., 2011).
* * *
2. **As mentioned above, the Methods section is very short, sometimes too short. For the interested reader more details on the methodology (e.g. on the CHIMERE model) should be provided either in an appendix, or in an SI section. In Section 3.3, more details on the optimization target and strategy need to be provided also in the main text (see details above).**

We fully agree with this point, which was fixed following general comments above.

3. **Figure 2 raises the question why the cumulative contributions of top and 4x side components are so stable over time, given the pronounced variability in contributions from different sides of the model domain, which is also discussed in Section 3.1? Maybe it would be helpful to show the mean concentrations from the global model that were taken as a reference for the 5 boundary values applied here.**

The stable contribution by boundary conditions is an optical illusion. There is a change in the y-axis at 1800 ppb making it look like the total of boundary conditions is flat, while it is not the case. There is a stretching by a factor 4 above and below 1800 ppb on the figure. The limit has been shifted to 1700 ppb to highlight the change in total boundary contribution and the legend has be amended accordingly.

As suggested by the reviewer, we also include an additional panel with the total simulations and observations only.

The legend reads as follows now:
* * *
(top panel) Observed and simulated total $CH_4$ concentrations; (middle panel) simulated contributions to total $CH_4$ concentrations; individual regions simulated by the model (see Fig. 1) are aggregated into two main continental components: North-America (NA) and Eurasia; light green areas depict Eurasian (mostly Siberian) wetlands, while dark green ones are North American wetlands. Shaded blue areas represent contributions from the sides of the CHIMERE simulation domain (see Fig. 1); orange shades represent minor anthropogenic contribution. So-called "top" line gives the simulated concentrations originating from the lower stratosphere (i.e., from the top of CHIMERE simulation domain). Please note the gap in y-axis scale at 1700 ppb highlighted by the dash line. (bottom panel) Observed and simulated isotopic ratios before (prior) and after (posterior) inversion.

4. **One confusing part of Section 3.2 is the mentioning of 'temporal variability' - what are the authors referring to? Obviously, there was no temporal variability in the source isotope ratios detected, or at least none is shown (or discussed) in any of the results. The only temporal variability shown is that of the retrieved isotope signal in the SWERUS data (Figure 2, bottom), but here the variability has been largely attributed to shifts in the origin of air masses, and accordingly source regions. The statement on temporal variability is repeated also in the conclusions. Please clarify.**

We agree that due to the unclear method parts, the mentioning of 'temporal variability' was misleading. We optimize the isotopic ratios of each contributions in the air masses sampled by SWERUS. Thus, for every time stamp of observation, posterior signatures are available. To keep consistent variability for each sector/region, temporal correlations were included as detailed in Table 1 and Eq.2-3.

The relevant amended part of the Method section are repeated below. We hope such clarifications will help the reader follow our approach easily.

In the end, we obtain hourly posterior signatures for each simulated sector and region for each of the 8000 inversions. Even though posterior signatures are available for each region and each sector at each observation time step, we do not inquire into the temporal variability of sources as constraints provided by the SWERUS observations are very heterogeneous in time and space. Instead, we compute overall posterior distributions for each simulated sector and region based on an ensemble of 8000000 (=8000 inversions $\times$ 1000 hourly observations). To minimizing the impact of control vector components that are ill-constrained by the inversion, all data points are not evenly counted in posterior distributions. Posterior distributions of signatures are computed accounting for all the Monte Carlo samples and weighted by the corresponding values of the sensitivity matrix **KH** (Cardinali et al., 2004), which gives an indicator of how much observations constrain one component of the control vector. The posterior optimal signature for each region and sector is computed as the maximum of the probability distribution.

5. **Finally, please make sure your reference list is up-to-date (old discussion papers cited)**

We updated papers that were still cited as discussion papers while accepted long ago. We try including the most recent publications on methane in the Arctic. We apologize for any missing reference not come to our knowledge.

6. **Some minor corrections include spelling and grammar errors. Some sentences do not make sense, e.g. the first sentence in the abstract. In addition, some further details of the how the model works could go into the supplementary information section. Figure 2 could be made clearer – it's a little hard to discern the observation line from the model results in the top panel. Maybe a separate panel with total model and observations would solve this. Also, the introduction would be stronger if it laid out current understanding of Arctic methane source types, thereby justifying the article by highlighting our current lack of understanding.**

We thoroughly spellcheck and proof-read the updated version of the manuscript to provide an English writing as good as possible. Figure 2 was made clearer as detailed above. We added references in the introduction to better set the scene.

**References**

[revised manuscript text omitted]